# Insights into Halogen-Induced Changes in 4-Anilinoquinazoline EGFR Inhibitors: A Computational Spectroscopic Study

**DOI:** 10.3390/molecules29122800

**Published:** 2024-06-12

**Authors:** Sallam Alagawani, Vladislav Vasilyev, Andrew H. A. Clayton, Feng Wang

**Affiliations:** 1Department of Chemistry and Biotechnology, School of Science, Computing and Engineering Technologies, Swinburne University of Technology, Melbourne, VIC 3122, Australia; salagawani@swin.edu.au; 2National Computational Infrastructure, Australian National University, Canberra, ACT 0200, Australia; vvv900@gmail.com; 3Optical Sciences Centre, Department of Physics and Astronomy, School of Science, Computing and Engineering Technologies, Swinburne University of Technology, Melbourne, VIC 3122, Australia

**Keywords:** tyrosine kinase inhibitors (TKIs), epidermal growth factor receptor (EGFR), absorption spectroscopy, time-dependent density functional theory (TD-DFT), ultraviolet-visible (UV-Vis)

## Abstract

The epidermal growth factor receptor (EGFR) is a pivotal target in cancer therapy due to its significance within the tyrosine kinase family. EGFR inhibitors like AG-1478 and PD153035, featuring a 4-anilinoquinazoline moiety, have garnered global attention for their potent therapeutic activities. While pre-clinical studies have highlighted the significant impact of halogen substitution at the C3’-anilino position on drug potency, the underlying mechanism remains unclear. This study investigates the influence of halogen substitution (X = H, F, Cl, Br, I) on the structure, properties, and spectroscopy of halogen-substituted 4-anilinoquinazoline tyrosine kinase inhibitors (TKIs) using time-dependent density functional methods (TD-DFT) with the B3LYP functional. Our calculations revealed that halogen substitution did not induce significant changes in the three-dimensional conformation of the TKIs but led to noticeable alterations in electronic properties, such as dipole moment and spatial extent, impacting interactions at the EGFR binding site. The UV–visible spectra show that more potent TKI-X compounds typically have shorter wavelengths, with bromine’s peak wavelength at 326.71 nm and hydrogen, with the lowest IC50 nM, shifting its lambda max to 333.17 nm, indicating a correlation between potency and spectral characteristics. Further analysis of the four lowest-lying conformers of each TKI-X, along with their crystal structures from the EGFR database, confirms that the most potent conformer is often not the global minimum structure but one of the low-lying conformers. The more potent TKI-Cl and TKI-Br exhibit larger deviations (RMSD > 0.65 Å) from their global minimum structures compared to other TKI-X (RMSD < 0.15 Å), indicating that potency is associated with greater flexibility. Dipole moments of TKI-X correlate with drug potency (ln(IC50 nM)), with TKI-Cl and TKI-Br showing significantly higher dipole moments (>8.0 Debye) in both their global minimum and crystal structures. Additionally, optical spectral shifts correlate with potency, as TKI-Cl and TKI-Br exhibit blue shifts from their global minimum structures, in contrast to other TKI-X. This suggests that optical reporting can effectively probe drug potency and conformation changes.

## 1. Introduction

The transmembrane receptor protein, epidermal growth factor receptor (EGFR), represents a pivotal target for pharmaceutical intervention, owing to its observed overexpression or mutations in non-small-cell lung carcinoma (NSCLC), and contributes significantly to the pathogenesis and progression of various other carcinoma types [1]. The integration of experimental and theoretical investigations at the molecular level serves as a reliable approach to comprehending the mechanisms underlying EGFR mutations. Such studies furnish essential insights for the design and development of novel protein tyrosine kinase inhibitors (TKIs). Notably, the discovery of potent TKIs exhibiting broad biopharmaceutical efficacy, including promising in vitro and in vivo antiproliferative activities, encompasses the identification of small molecule inhibitors referred to as “tyrphostins” [2].

Tyrosine kinase inhibitors (TKIs) belonging to the 4-quinazolinamine class feature a quinazoline scaffold, which is widely prevalent among both synthetic and natural bioactive compounds [3]. Derivatives containing quinazoline constitute a significant category of synthetic compounds and serve as a compelling scaffold for the development of anticancer drugs. Over the years, these compounds have captured attention due to their diverse biological activities, particularly as kinase inhibitors, contributing to their prominence in research and drug design [4,5,6,7]. Bridges and coworkers [3] have notably synthesized an impressive series of quinazolines to elucidate the structure-function relationships inherent in this drug class.

The information learned from the structural activity relationship (SAR) investigation of 4-anilinoquinazoline derivatives is that a phenyl ring attached via NH linkage at the C4 position is an important determinant of drug efficacy. Notably, within this phenyl ring, the substitution of -Br at the C3 position demonstrated superior potency in inhibiting the EGFR compared to other substitutions, such as H. The noteworthy observation that a series of halogen substitutions could effectively modulate the potency of this drug class was particularly intriguing to us. Inhibition constants, expressed as IC_50_ values of 29, 3.8, 0.31, 0.025, and 0.89 nM for hydrogen, fluorine, chlorine, bromine, and iodine, respectively, were measured by varying the halogen at position 3 of the aniline ring, with the Br derivative emerging as the most effective inhibitor [3]. This leads to the question: can an understanding of changes in potency be achieved by examining the structure and properties of the free molecule?

Halogens are of significant importance in various natural phenomena and the creation of synthetic materials, primarily because of their distinctive physicochemical characteristics and their ability to participate in a wide range of diverse interactions [8]. Moreover, halogens have a pivotal role in intricate biological mechanisms, including processes like ligand binding or molecular folding.

The second motivation comes from the fact that quinazolines, characterized by aromatic conjugated ring systems, are also fluorescent when excited with UV light. Optical spectra of the quinazoline-based TKIs are sensitive to interaction with the environment, with a particular sensitivity to parameters such as hydration and solvent polarity [9].

Recently, Wang and co-workers established experimentally validated theoretical methods to study optical spectra properties of one of the most well-studied and highly potent TKIs, i.e., Tyrphostin AG-1478 (N-(3-chlorophenyl)-6,7-dimethoxy-4-quinazolinamine) [10,11,12]. The experimentally measured UV-vis and fluorescence spectra of AG-1478 [10] can be employed to validate and benchmark quantum mechanical methods such as density functionals in density functional theory (DFT) [13], in order to identify a small group of excellent-performing DFT functionals to study other TKIs.

However, TKIs built on the 4-anilino-quinazoline scaffold exhibit flexibility, adopting various structural forms, and they tend to form groups of these conformations, known as conformation clusters or ensembles, when placed in solvents or when interacting with target kinases. In ligand-based drug design, finding a favourable drug conformation with low energy, often achieved through an extensive search for different conformations, becomes critically important, especially when there is limited or no available structural information about the target [14]. The recognition of the importance of identifying low-energy conformations, commonly referred to as preferred conformations, has been a well-established concept, particularly in the quest for bioactive conformations, known as active conformers.

Developing reliable, scalable, and universally applicable techniques for ascertaining the stereochemistry of pliable organic substances like pharmaceuticals has been a challenging endeavour. This challenge is particularly pronounced when dealing with all TKIs that possess the quinazoline-aniline scaffold, as they inherently exhibit multidimensionality and flexibility within a three-dimensional (3D) space. As a result, accurately and consistently identifying the most favoured (low-energy) conformations while adhering to a predefined level of strain energy remains an elusive goal [12].

In this research, we employ quantum-mechanical simulations of optical spectra to provide insights into the contribution of conformational variations in TKIs-X in DMSO (Dimethyl Sulfoxide) solvent. The investigation also sheds light on how the presence of halogens (fluorine, chlorine, bromine, and iodine) influences the arrangement of low-energy conformations and their associated optical spectra of EGFR database crystal structures and TKIs-X.

## 2. Results

### 2.1. Conformation Changes in TKI-Xs Due to Halogen Substitution and Strain Energy (SE)

Constraining molecular conformation can be employed to enhance the comprehension of particular interactions between drugs and receptors. Even in the scenario where a medication substance selectively attaches to a singular receptor, there are still potential benefits associated with utilizing conformationally constrained analogs. Constraining conformational flexibility aids in the mapping of receptor-binding sites and has the potential to yield more potent compounds [15]. Hence, conformations of TKIs-X, where X represents hydrogen or halogens, will be investigated in a DMSO solvent to elucidate the influence of halogens on the conformational distribution of TKIs.

The IUPAC name of TKI-H is 6,7-dimethoxy-N-phenylquinazolin-4-amine, of TKI-F is N-(3-fluorophenyl)-6,7-dimethoxyquinazolin-4-amine, of TKI-Cl (AG1478) is N-(3-chlorophenyl)-6,7-dimethoxyquinazolin-4-amine, of TKI-Br (PD153035) is N-(3-bromophenyl)-6,7-dimethoxyquinazolin-4-amine, and of TKI-I is N-(3-iodophenyl)-6,7-dimethoxyquinazolin-4-amine. The nomenclature is additionally provided in Appendix A. 

Accurate determination of electronic states in conformers with low energy could be achieved exclusively through the application of quantum-mechanical techniques, which were utilized to compute the molecular characteristics of TKIs-X conformers, along with their UV–visible spectra. The forthcoming discourse will primarily focus on the four most energetically favourable local minimum conformers, each with an energy level of approximately SE < 3.0 kcal mol^−1^. The determined lowest-energy (global-minimum) conformation of TKI-H, TKI-F, AG1478, PD153035, and TKI-I, in a DMSO solvent environment were characterized with total energies −933.798880 E_h_, −1033.081728 E_h_, −1393.425183 E_h_, −3507.379134 E_h_, and −1230.996026 E_h_, respectively (see Appendix A), using B3LYP/def2TZVP model in DMSO solvent. 

Strain energy (SE) is the excess energy of a conformer above its global minimum structure of an inhibitor. Figure 1 depicts the observed pattern in the strain energies (SEs) of the five low-energy conformations within the DMSO solvent for each TKI-X, in comparison to their corresponding individual global minimum structure, along with the structures retrieved from the EGFR database (http://crdd.osdd.net/raghava/egfrindb/, accessed on 2 February 2023). The energies of the final structures (EGFR-DB) were optimized with the torsion angles held constant [16]. Furthermore, Figure 1 illustrates the considerable diversity in SE for the EGFR database structures. For example, the SE is given by 8.193 kcal/mol for TKI-Cl (AG1478) and 8.279 kcal/mol for TKI-Br (PD153035). In contrast, there is a relatively minor increase in SE for TKI-H, TKI-F, and TKI-I, amounting to 1.346, 0.876, and 1.040 kcal/mol, respectively. 

Table 1 compares molecular properties between the global minimum conformer and the EGFR database structures of TKIs-X, determined through calculations using the same DFT B3LYP/def2TZVP model in DMSO solvent. The distribution of the population is influenced by the degeneracy of conformational states. For instance, at room temperature, TKI-H conformer 1 manifests approximately 89.7% of the population, while TKI-F shows a population of 61.2% for conformer 1. TKI-Cl (AG1478) exhibits a population of 63.5% for conformer 1, TKI-Br (PD153035) shows 62.7%, and TKI-I displays a population of 54.6% for the same conformer. 

The dipole moments (DMs) of the global conformers **1** seem associated with the TKI potency. The order of the dipole moments of the TKI-Xs is the same as the order of the potency, the IC_50_ values in (nM) [3]. For example, the dipole moments of TKI-H, F, Cl, Brand I are 5.80D, 7.79D, 8.08D, 8.23D, and 7.99D, corresponding with the measured IC_50_ values of 29 nM, 3.8 nM, 0.31 nM, 0.025 nM, and 0.89 nM, respectively [3]. Figure 2 reports the relationship between dipole moments (DMs) and potency (ln(IC_50_ in nM). The relationship between the dipole moment and electronegativity typically aligns in a manner where electronegativity tends to lead to larger dipole moments. Following this, elements like fluorine, chlorine, and bromine, which boast higher electronegativities compared to hydrogen, display an expected increase in their dipole moments. However, the observed decreases in the dipole moment for iodine, despite its larger size, could potentially be attributed to a multifaceted interplay of factors. This may include a reduction in electronegativity relative to smaller halogens, coupled with probable electronic influences within the molecular structure, which might counteract the anticipated rise in the dipole moment linked with size and passing through the aromatic 4-anilinoquinazoline conjugate system. Furthermore, alterations in atomic polarizability, signifying the ease of electron cloud distortion under an external electric field, could also contribute to the atypical trend noticed in the dipole moments of the conformers. 

Additional properties of the various conformational forms are collected in Appendix A. It should be noted that the halogen series, conformers 1, 3, and 5, exhibit comparatively higher dipole moments, measuring around 8D, 7D, and 9D, respectively. In contrast, conformers 2 and 4 demonstrate notably smaller dipole moments, approximately around 4.5D. The increased dipole moment of configuration 5 could potentially enhance the favourable binding interaction with the EGFR.

The specific properties of the top five conformers of TKIs-X in the DMSO solvent are collected in Appendix A. The global minimum structure **1** for each TKIs-X exhibits planar structures with a 3-X-Ph orientation; in the case of conformers **2**, an in-plane rotation (rotational isomerism) has occurred, resulting in a 180-degree flip of the phenyl ring containing the X atom (5-X-Ph orientation). Conformer **3** maintains a planar configuration; however, there is a specific rotation within the plane for each TKIs-X. When X is a halogen, this rotation involves the oxygen atom (atom 28) of the methoxy group. Conversely, when X is hydrogen, the conformer transitions to a non-planar state, and the alteration involves the rotation of the phenyl ring into the same plane. Conformer **4** exhibits similar changes in configuration as observed in conformer **3**, along with an in-plane rotation leading to a 180-degree flip of the phenyl ring (5-X-Ph orientation) for each TKIs-X when X is a halogen. However, when X is hydrogen, the conformer transitions to a non-planar state, and the alteration involves the rotation of the phenyl ring out of the same plane. Conformer **5** presents a contrasting scenario compared to conformer **3**; it retains a planar structure, yet there is a distinct rotation within the plane for each TKIs-X when X is a halogen or hydrogen, and this rotation affects the other oxygen atom (atom 27) of the methoxy group. 

It is known that a potent inhibitor often takes a higher energy conformer rather than the global minimum structure. The alignment of the crystal structure and the global minimum of each TKI-X can be estimated using root mean square deviation (RMSD) (Å) values. The RMSD indicates how different the crystal structures of the TKI-X are from its global minimum structure. As a result, the TKI-X configurations from the EGFR database (http://crdd.osdd.net/raghava/egfrindb/, accessed on 2 February 2023) are aligned with the global minimum conformers **1** of TKIs-X. All configurations are arranged in a coordinated manner based on the plane constituted by three atoms within the quinazoline core: C_(1)_, C_(4)_, and C_(10)_ [12]. The deviations in conformation commence at the two methoxy (-OCH_3_) side appendages situated on C_(1)_ and C_(6)_, associated with the methoxy (-OCH_3_) groups. Subsequently, there is a reorientation of the X-phenyl ring observed in TKI-Cl and TKI-Br. The RMSD values of the TKI-X are 0.135 Å, 0.121 Å, 0.657 Å, 0.692 Å, and 0.124 Å for TKI-H, YKI-F, TKI-Cl, TKI-Br, and TKI-I, respectively. It is interesting to know that the crystal structures of the more potent TKI-Cl and TKI-Br are more distorted from their respective global minimum structures.

### 2.2. Changes in the UV–Visible Spectra of the TKIs-X Conformers

The absorption profile of a compound in the UV-Vis spectrum originates from the transition that takes place between the molecular orbitals that are filled (occupied) and those that are unfilled (virtual) [18]. The conjugated ring system present in 4-anilinoquinazoline TKIs-X renders them highly suitable for discovering their ultraviolet-visible (UV-Vis) spectral properties [19]. Figure 3 depicts the TD-DFT calculated UV–visible absorption spectrum spanning from a wavelength exceeding 320 nm to 340 nm for EGFR-DB in a DMSO solvent. As depicted in the spectra, the prominent spectral band with wavelengths exceeding 320 nm can be attributed to a robust, singular electronic transition. For instance, the transitions occurring at specific wavelengths, such as 333.17 nm with an oscillator strength of 0.62 in hydrogen, 330.98 nm with an oscillator strength of 0.64 in fluorine, 327.54 nm with an oscillator strength of 0.53 for chlorine, 326.71 nm with an oscillator strength of 0.53 for bromine, and 332.84 nm with an oscillator strength of 0.67 for iodine, are primarily dictated by the highest occupied molecular orbital to lowest unoccupied molecular orbital (HOMO-LUMO) transitions, each contributing substantially at 49%. In the UV-Vis spectra analysis, it is observed that the TKI-X, which is more potent, exhibits shorter wavelengths and a blueshift. For instance, when comparing the IC_50_ values (nM), bromine with an IC_50_ of 0.025 nM shows its maximum peak at 326.71 nm. Following this, chlorine with an IC_50_ of 0.31 nM has its peak wavelength at 327.54 nm. Conversely, the hydrogen compound, which has the lowest IC_50_ value of 29 nM, displays a rightward 7 nm red shift in its peak wavelength, measuring at 333.17 nm. Moreover, similar UV–visible wavelength peaks and spectra are observed between 210 and 330 nm for bromine and chlorine, which possess higher potencies compared to other TKIs-X with lower potency values. Refer to Appendix A for the comprehensive UV–visible spectra of EGFR-DB in a DMSO solvent. 

The primary transitions, featuring an oscillator strength ƒ > 0.22 and electronic transition configurations with a contribution of more than 5%, are specified in Appendix A for EGFR-DB structures and Appendix A for the global minimum conformers in the Appendix A. Additional spectral bands observed at wavelengths below 320 nm are the result of numerous transitions involving frontier orbitals with low energy, like HOMO-n and LUMO+m, with n and m representing integers within a limited range (1, 2, 3, …, 9), indicating transitions associated with lower energy levels. The exact positions of the spectral bands in this range cannot be ascertained through the consideration of individual transitions alone. Instead, they require the utilization of the same fitting program employed in experimental measurements. Nonetheless, we have provisionally determined the positions by identifying the peaks of the highest points. For a comparative analysis of the UV-Vis absorption spectral maxima between the EGFR-DB and the five lowest energy conformers of TKIs-X, consult Appendix A in the Appendix A.

Additionally, consult Appendix A for a comparative analysis of the UV-vis spectra of EGFR-DB. Notably, the structural resemblance among these compounds, characterized by sharing the common base of 6,7-dimethoxy-N-phenylquinazolin-4-amine, except for the variance in the atom (24) of the phenyl ring, contributes significantly to their spectral similarity. Meanwhile, differences observed in the spectra are attributed to the effects stemming from the substitution of halogens at the atom (24). For example, there are distinctions at the onset of the spectra. For instance, the iodine spectrum is more confined, commencing after 200 nm and concluding around 340 nm, while fluorine and hydrogen spectra commence at 180 nm, chlorine at 190 nm, and bromine at 195 nm. The key factors differentiating halogens and their effects on UV-vis spectra include their atomic structure, electronic configuration, and electronegativity. Variations in UV-Vis spectra among halogens are attributable to their unique electronic structures [20]. This can manifest in discrepancies such as the intensity and position of absorption peaks and the range of wavelengths at which transitions occur, which might vary due to the different electronic configurations and bonding interactions involving halogens. The specific spectral transitions and features observed are influenced by the individual electronic properties of the halogen atoms and their interactions within the molecular environment.

The potency of the TKI-X inhibitors seems to relate to the absorption optical shift. Table 2 compares the optical properties with respect to the calculated maximum absorption wavelength (λ_max_) of the global minimum structures **1** and EGFR-DB of the TKIs-X with the oscillator strengths, ƒ ≥ 0.5. The substitution of X leads to optical spectral shifts (λ_max_) from the crystal structure to the global minimum structures **1**, depending on the X atoms on the TKI. When X becomes hydrogen (H), fluorine (F), and iodine (I), the λ_max_ of their optical spectra of TKI-X exhibits a small red shift with 1.96 nm, 1.13 nm, and 1.33 nm, respectively. In contrast, TKI-X with X as chlorine (Cl) and bromine (Br) show redshift as −2.34 nm and −3.28 nm, respectively (highlighted in the table). Interestingly, Cl and Br show a blue shift from their EGFR-DB spectrum compared to the calculated global minimum structure spectrum, while the others exhibit a red-shift peak. This finding agrees with prior studies conducted using DFT-B3LYP. In our study, the absorption peak wavelengths of the global minimum structures for Br and Cl are 329.99 and 329.88, respectively, compared to 331.77 for Br and 331.02 for Cl in the previous study [16,19]. 

Figure 4 presents the correlation between optical shift and the potency of the TKI-X. As shown in the figure, the more blue shifts in their optical spectra (the blue columns in the figure) between the global minimum structures and their corresponding crystal structures from the EGFR database, the more potent it seems to be. Note that the relationship is more obvious when using the natural logarithm function ln(IC_50_) function, as the potency indicator is usually very small. 

Figure 5 compares the entire UV-Vis absorption spectrum in a DMSO solvent ranging from 180 to 350 nm for the more potent TKIs, (a) TKI-Br and (b) TKI-Cl, which exhibit a different pattern compared to the others, H, F, and I. It presents spectra of the calculated global minimum structure, EGFR-DB structure, and the UV-Vis difference spectra produced by subtracting the spectrum of the calculated global minimum with the UV-Vis spectrum of the EGFR-DB (λcal−λEGFR-DB) for both TKI-Br and TKI-Cl (dash spectra). In Figure 5a for bromine, the transparent-red spectrum represents TKI-Br with a peak absorption at 326.71 nm, and the red spectrum represents the calculated global minimum structure of TKI-Br with a blue-shifted peak at 329.99 nm. The figure illustrates that in the lower transition bands within the wavelength region of 180–230 nm, the transparent-red UV-Vis spectrum of TKI-Br shows one sharp peak with a strong absorption intensity at 220 nm shifted slightly from the 216 nm in the red spectrum, contrasting with the two moderate peaks observed in the red UV-Vis spectrum of the calculated global minimum structure of Br at 199 nm and 216 nm. Additionally, the onset of the red spectrum occurs at 199 nm, while for the transparent-red spectrum, the onset exhibits a blue shift at 201 nm. Difference absorbance spectra presented in a dashed line show the highest difference value between the calculated and EGFR spectra of TKI-Br occurred at 222 nm position bands, following the difference peak at 324 nm. In Figure 5b, the transparent-green spectrum represents TKI-Cl, showing its highest absorption peak at 327.54 nm. In contrast, the green spectrum depicts the calculated global minimum structure of TKI-Cl, which has a slightly shifted peak at 329.88 nm. Within the wavelength range of 180–230 nm, the transparent-green UV-Vis spectrum of TKI-Cl displays two sharp peaks with strong absorption at 196 nm and 220 nm, contrasting with the two moderately intense peaks observed in the green UV-Vis spectrum at 197 nm and 216.5 nm. Moreover, the transparent-green spectrum begins its absorption at 189 nm, while the green spectrum starts at a slightly higher wavelength of 195 nm. The difference absorbance spectra highlight the greatest disparities between the calculated and EGFR spectra of TKI-Cl at the positions of 222 nm, 197 nm, and 247 nm, following the difference peak at 324 nm.

The λmax transition of the absorption spectra of the TKI-X are transitions between the highest occupied molecular orbital (HOMO) and the lowest unoccupied molecular orbital (LUMO). The LUMO-HOMO for TKI-Cl are orbital 82 and orbital 83, respectively, and for TKI-Br, are orbital 91 and orbital 92, respectively. Figure 6 presents the natural transition orbital (NTO) method performing separate unitary transformations on occupied and virtual MOs, resulting in only one or a few dominant orbital pairs with significant contributions. 

Natural transition orbitals (NTOs) illustrate the nature of optically active singlet excited states in the absorption bands, which help to recover an electron/hole picture while preserving phase information [21,22]. Specifically, the compound (a) TKI-Cl is examined for the NTO82→NTO83 with an oscillation strength of 0.5732 and the compound (b) TKI-Br for the transition of NTO91→NTO92 with an oscillation strength of 0.5542. As can be seen, although there are some similarities in the HOMOs and LUMOs of the TKI-Cl and TKI-Br, the electron distributions of the potent TKIs are not the same, reflecting the halogen impact. For example, the HOMO of TKI-Cl (NTO81) and the HOMO of TKI-Br (NTPO91) exhibit major differences in the quinazoline moieties and the phase where the HOMO (NTO91) of TKI-Br is more delocalized than the HOMO (NTO82) of TKI-Cl.

The dominant NTO pairs for the first four excited states of (a) TKI-Br and (b) TKI-Cl are shown in Figure 7. The first excited state is at the top of the figure, the “hole” is on the left, and the “particle” is on the right for each compound and state. The associated eigenvalues λ for TKI-Br are 0.982423, 0.990344, 0.730883, and 0.861056, respectively. For (b) TKI-Cl, the associated eigenvalues λ are 0.983168, 0.980275, 0.729861, and 0.896348, respectively. This eigenvalue is >0.98 for the first two excited states, indicating that even the heavily mixed second state can be accurately described by a dominant excitation pair, which accounts for over 98% of the transition “percentage contribution to the electron excitation”. 

## 3. Computational Details

In this study, all calculations are the same as described in [12] for TKI-Cl which established the conformational search method, except for the basis set, which is def2TZVP as the basis set in [12] is not available for iodine. The solvent is the same dimethyl sulfoxide (DMSO). Our comprehensive random search method provides a means to rapidly generate conformational distributions of complex molecules including conformers with significant stain energy. It should be noted that this type of method generates minima which do not necessarily correspond to the true global minimum. Global minima configuration methods (stochastics or genetic algorithms) provide a more exhaustive search in conformational space at a substantially higher cost.

The optimized five low-lying TKIs-X conformer clusters are each associated with a distinct local minimum, with the lowest one being the global minimum structure of the TKI-X, as displayed in Appendix A, specifically, in Appendix A. The UV-Vis spectra of the five low-lying conformers for each TKIs-X were computed in a DMSO solvent, utilizing the time-dependent density functional theory (TD-DFT) method in conjunction with the B3LYP/def2TZVP model. UV-vis absorption spectra were determined for each TKIs-X, considering the local minimum conformations and taking into account the lowest 30 singlet excited states. All computational simulations were conducted utilizing the computational chemistry software Gaussian 16 [23] within the computational resources provided by the Swinburne OzSTAR and Ngarrgu Tindebeek supercomputing facilities.

## 4. Conclusions

This study focuses on the impact of halogen on the EGFR TKI-X where X = H, F, Cl, Br, and I on the C3’ position of the aniline-quinazoline scaffold. The four lowest-lying conformers of each TKI-X, together with their crystal structure from the EGRF database, are studied and compared. The present study confirmed that the most potent conformer of the inhibitors is often not the global minimum structure but one of the low-lying conformers. As a result, the conformation sampling developed in [12] is very useful without missing any potent conformation of the inhibitors. The more potent TKI-Cl and TKI-Br exhibit larger deviations (RMSD > 0.65 Å) from their global minimum structures and the crystal structures, as the calculated RMSD (Å) of these TKIs are larger than other TKI-X (RMSD < 0.15 Å), indicating these TKIs (H, F and I) are less flexible. Next, the dipole moments of the TKI-X are related to the drug potency in its nature logarithm function ln(IC50 nM). The TKI-Cl and TKI-Br in their global minimum structures and their crystal structures exhibit large dipole moments of μ > 8.0 Debye, whereas the dipole moments of other TKI-X (H, F, and I) are significantly smaller. Perhaps the most significant correlation of the TKIs is the optical spectral shifts and the natural logarithm function of the potency, although the relationship is not linear. That is, the more potent TKI-Cl and TKI-Br exhibit apparent optical blue shifts between the global minimum structures and the crystal structures. The optical shift for TKI-Cl and TKI-Br are blue-shifted from their crystal structures, which is opposite to the other TKI-X (H, F, and I). As a result, optical reporting effectively probes the drug potency and conformation changes. 

It is important to emphasize that the drug-protein interaction is also dynamic. The potential integration of quantum mechanics (QMs) TKIs search combined with molecular dynamics (MDs) simulations is anticipated to offer a powerful approach for studying TKIs’ binding with the proteins of the epidermal growth factor receptor (EGFR) target in future investigations.

## Figures and Tables

**Figure 1 molecules-29-02800-f001:**
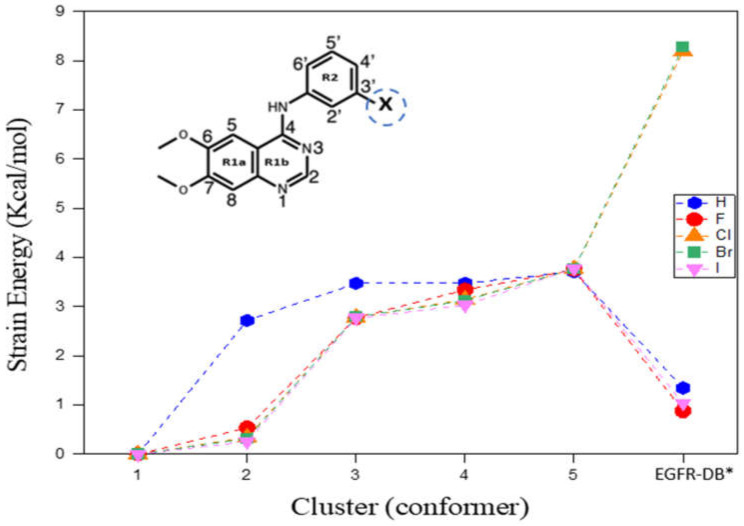
Strain energies (SEs) of the top five low-energy TKIs-X conformers calculated in DMSO solvent. * EGFR-DB structures were obtained by downloading them from the EGFR database (http://crdd.osdd.net/raghava/egfrindb/, accessed on 2 February 2023) (H: EGIN0000732, F: EGIN0000733, Cl: EGIN0000281, Br: EGIN0000010, and I: EGIN0000736) [17], after which optimization was conducted while maintaining fixed torsion angles. The enclosed structure of TKIs-X presents the IUPAC nomenclature for the compound.

**Figure 2 molecules-29-02800-f002:**
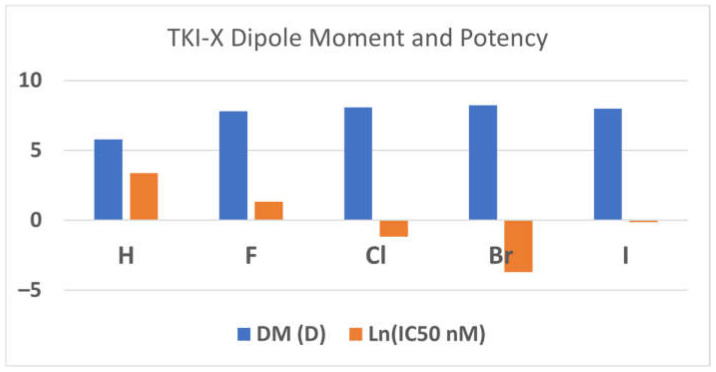
The relationship between DM (in Debye) and the inhibitor potency (ln(IC_50_ in nM)). Note that the more negative the lnX function, the smaller X, so the more potent.

**Figure 3 molecules-29-02800-f003:**
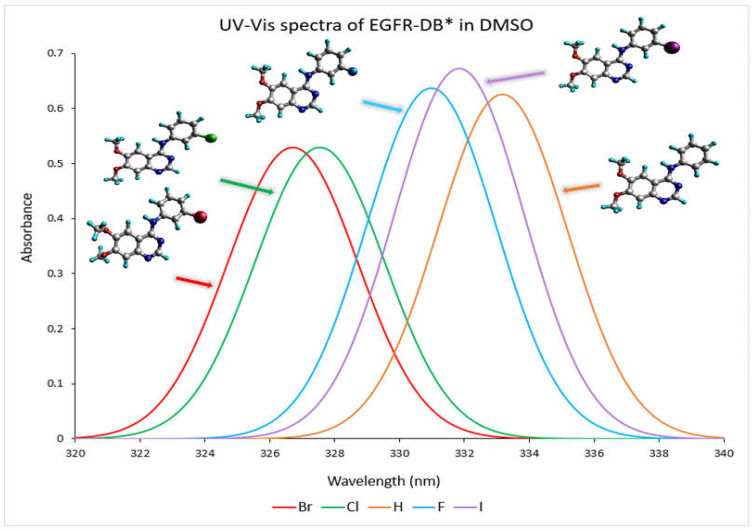
The UV–visible absorbance spectra (nm) of the EGFR* database crystal structures in DMSO solvent were determined through the DFT B3LYP/def2TZVP computational method. The spectral colours corresponding to each structure are as follows: H-EGIN0000732 is represented by an orange spectrum, F-EGIN0000733 by a blue spectrum, Cl-EGIN0000281 by a green spectrum, Br-EGIN0000010 by a red spectrum, and I-EGIN0000736 by a purple spectrum. * EGFR database source is (http://crdd.osdd.net/raghava/egfrindb/, accessed on 2 February 2023) [17].

**Figure 4 molecules-29-02800-f004:**
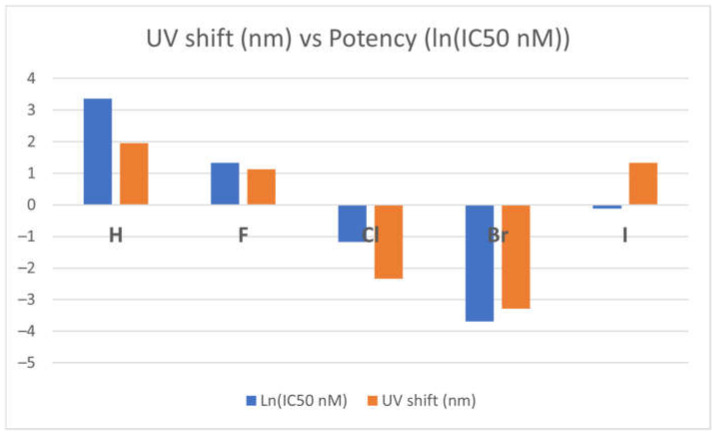
Correlation between optical shift and the potency of the TKI-X.

**Figure 5 molecules-29-02800-f005:**
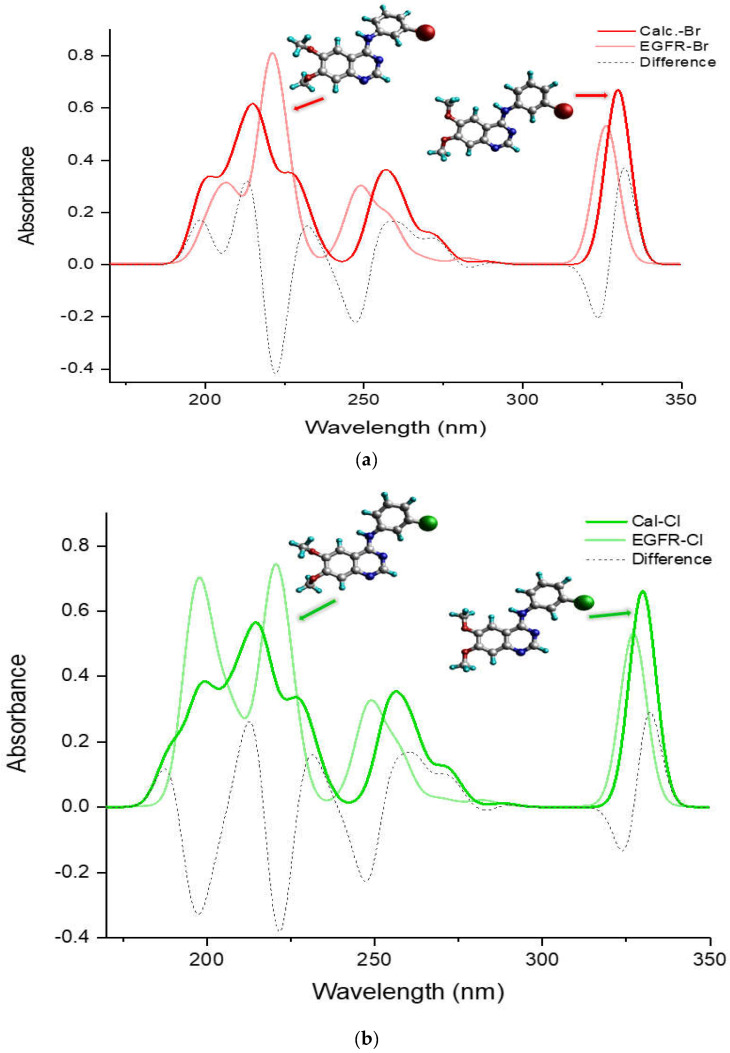
Comparison of UV-Vis absorption spectra (nm) of the calculated global minimum structure (red) and EGFR-DB [17] structure (blue) with their UV-Vis differential spectrum produced by (λcal−λEGFR-DB) (green) for (**a**) TKI-Br, (**b**) TKI-Cl, using B3LYP/def2TZVP method in DMSO solvent.

**Figure 6 molecules-29-02800-f006:**
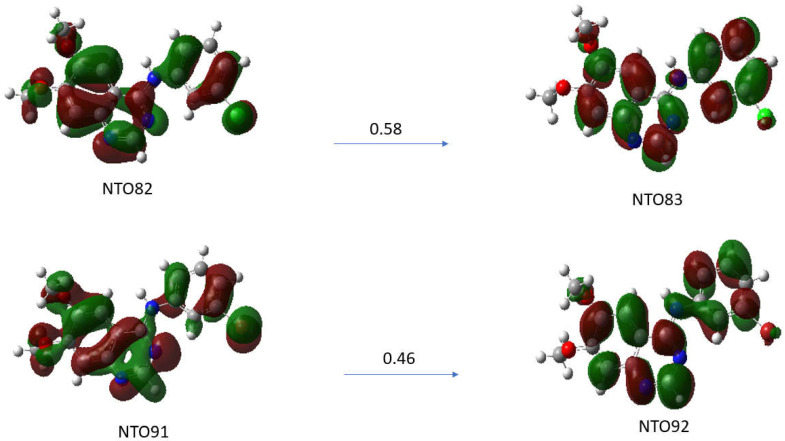
Natural transition orbitals (NTOs) analysis for the excited state with higher oscillation strength for compounds (**a**) TKI-Cl for excited state S0 S15 (ƒ = 0.5732), (**b**) TKI-Br for excited state S0 S17 (ƒ = 0.5542). Calculated using TD-DFT at the B3LYP/def2TZVP level of theory in DMSO solvent.

**Figure 7 molecules-29-02800-f007:**
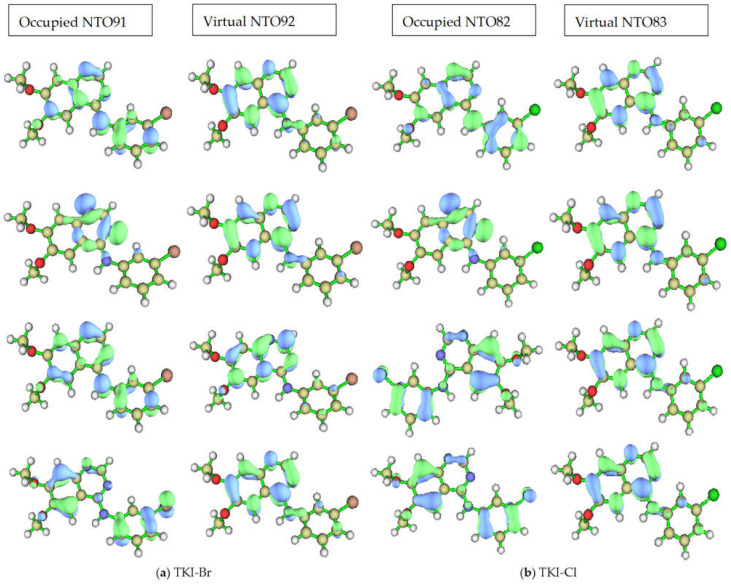
The dominant NTOs (occupied and virtual) pairs for the first four excited states of (**a**) TKI-Br. The associated eigenvalues λ are 0.982423, 0.990344, 0.730883, and 0.861056, respectively; (**b**) TKI-Cl. The associated eigenvalues λ are 0.983168, 0.980275, 0.729861, and 0.896348, respectively.

**Table 1 molecules-29-02800-t001:** Comparison of molecular properties of the global minimum conformers **1** and EGFR database crystal structures of TKIs-X in DMSO solvent.

X	Property	H	F	Cl	Br	I
Global minimum	*E* (E_h_) ^#^	−0.798880	−0.081728	−0.425183	−0.379134	−0.996026
(1)	μ (Debye)	5.8286	7.7915	8.0755	8.2320	7.9890
	%	89.665	61.211	63.493	62.711	54.619
	Structure	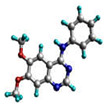	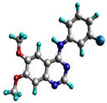	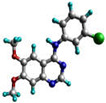	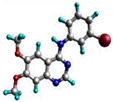	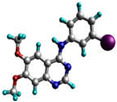
EGFR-DB *	Δ*E* (Kcal mol^−1^)	1.346	0.876	8.193	8.279	1.040
	μ (Debye)	5.8063	5.9049	0.7406	2.4071	5.8799
	%	8.880	13.603	0.001	0.001	9.156
	Structure	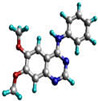	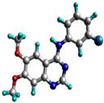	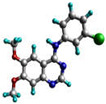	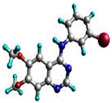	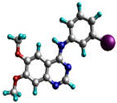
	RMSD (Å)	0.135	0.121	0.657	0.692	0.124

* EGFR database crystal structure (http://crdd.osdd.net/raghava/egfrindb/, accessed on 2 February 2023) (H: EGIN0000732, F: EGIN0000733, Cl: EGIN0000281, Br: EGIN0000010, and I: EGIN0000736) [17]. Here Δ*E* is the energy of the TKIs-X above its corresponding global minimum structures. ^#^ The total energies in E_h_ of the global minimum structures of TKIs-X need to add −933, −1033, −1393, −3507, and −1230 E_h_, respectively, for X = H, F, Cl, Br, and I. All calculations use the DFT B3LYP model revealed in Computational Details.

**Table 2 molecules-29-02800-t002:** Comparison of maximum absorption band shifts in the calculated global minimum and EGFR-DB structures of TKIs-X and their oscillator strengths (ƒ > 0.5) ^a^.

X	Calc. Global Minimum		EGFR-DB *		Δλ_max_ (nm)	Ln(IC_50_ (nM))
	λ_max_ (nm)	ƒ	λ_max_ (nm)	ƒ		
H	331.21	0.6391	333.17	0.6249	1.96	3.37
F	329.85	0.6462	330.98	0.6369	1.13	1.33
Cl	329.88	0.6611	327.54	0.5294	−2.34	−1.17
Br	329.99	0.6710	326.71	0.5295	−3.28	−3.69
I	330.51	0.6853	331.84	0.6728	1.33	−0.11

^a^ Δλ_max_ = λcrystal − λcal, in nm. Calculations use B3LYP/def2TZVP method in DMSO solvent. λ: denotes excitation energy, ƒ: represents oscillator strength. * EGFR database (http://crdd.osdd.net/raghava/egfrindb/, accessed on 2 February 2023) (H: EGIN0000732, Cl: EGIN0000733, F: EGIN0000281, Br: GIN0000010, and I: EGIN0000736) [17].

## Data Availability

Data are contained within the article and Appendix A.

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
