# Peer review of "Insights into Halogen-Induced Changes in 4-Anilinoquinazoline EGFR Inhibitors: A Computational Spectroscopic Study"

_molecules, 2024, doi:10.3390/molecules29122800_

Round 1

Reviewer 1 Report

Comments and Suggestions for Authors

This paper investigates the effect of halogen substitutions on the structure of tyrosine kinase inhibitors.  The paper is the first step toward further research employing docking or molecular dynamics simulations on these systems.  The paper is well-written and explained. The work should be published with minimal adjustments.  

(1)  The authors' comprehensive random search cannot guarantee that the minima energy corresponds to the true global minima. Global minima configuration methods (stochastics or genetic algorithms) provide a more exhaustive search in conformational space at a substantially higher cost. I propose noting this in the paper.

(2)  In line 408-411 the authors say: “The UV-visible spectra indicate that more potent TKI-X compounds typically have shorter wavelengths, with bromine, chlorine, and least potent hydrogen demonstrating distinct shifts in lambda max, correlating with their respective IC50 values in (nM).” I agree that wavelengths follow qualitatively the trends in IC50, but is there a quantitative relationship between these numbers?

Reviewer 2 Report

Comments and Suggestions for Authors

The manuscript entitled “Insights into Halogen-Induced Changes in 4-Anilinoquinazoline EGFR Inhibitors: A Computational Spectroscopic Study” submitted by S Alagawani and coworker presents a detailed theoretical investigation about the conformational structure and UV absorption properties of the halogen substituted tyrosine kinase inhibitors (TKI) drug molecule using the DFT and TD-DFT frameworks. The research topic explored in the manuscript looks of having a significant importance since it is a systematic and exhaustive investigation about the role of the halogen substitution on the TKI’s conformational changes and on its optical properties. The manuscript is well written and contains an interesting methodological discussion and analysis. The molecular modelling techniques have been applied at the highest scientific level, and the results obtained with these methods are of great interest.

General observation:

- Although, the analysis of both conformational and optical activity has been discussed in detail and exhaustively, these two types of computation alone can be considered to be routinely used. Especially in the light of the fact that the optical property of halogen-free TKI has been extensively discussed. See: Ref-s 9-13, 16 and 19. In order to understand halogen substitution in more detail, the nature of the electron transitions would also need to be better understood. For example, a discussion of the Natural Transition (or Difference) Orbitals would make the nature of these transitions much clearer. See e.g. Phys. Chem. Chem. Phys., 2024, 26, 3755-3794.

- If one wants to understand photochemical behavior, it is essential to follow the relaxation of the S1 excited state, i.e. the fluorescence behavior and the effect of the halogen substituent on it. It would be interesting to see whether the presence of different halogen atoms affects this relaxation.

- The EGFR database (http://crdd.osdd.net/raghava/egfrindb/) link cannot be open.

- Taking dispersion effects into account (For ex. considering the Grimme’s D3 empirical correction), even if it is only a monomeric configuration, is very important because these types of interactions within the molecule can affect the conformational energy differences and barriers.

I believe that the manuscript would be suitable for publication in Molecules, but it needs considerable revision and improvement.

Round 2

Reviewer 2 Report

Comments and Suggestions for Authors

This new revised version of the manuscript shows significant improvements compared to the first submitted version of the manuscript. The author has given satisfactory solutions and explanations for the addressed questions related to his original work.